# CONSTITUENCY TREE REPRESENTATION FOR ARGUMENT UNIT RECOGNITION

## ABSTRACT

The extraction of arguments from sentences is usually studied by considering only the neighbourhood dependencies of words. Such a representation does not rely on the syntactic structure of the sentence and can lead to poor results especially in languages where grammatical categories are scattered in the sentence. In this paper, we investigate the advantages of using a constituency tree representation of sentences for argument discourse unit (ADU) prediction. We demonstrate that the constituency structure is more powerful than simple linear dependencies between neighbouring words in the sentence. Our work was organised as follows: First, we compare the maximum depth allowed for our constituency trees. This first step allows us to choose an optimal maximum depth. Secondly, we combine this structure with graph neural networks, which are very successful in neural network tasks. Finally, we evaluate the benefits of adding a conditional random field to model global dependencies between labels, given local dependency rules. We improve the current best models for argument unit recognition at token level and also use explainability methods to evaluate the suitability of our model architecture.

## 1 INTRODUCTION

Argument Identification inside documents is the first step to study rhetorical speech processes or political debates. It is a concrete NLP application that interests the media community during the electoral periods.

The goal is to precisely identify arguments discourse unit (ADU) (defined as the minimal analysis units) in a sentence and predict its stance, given a topic.

Former works on argument analysis at the token level Trautmann (2020) has used language models such as the Bidirectional Encoder Representations from Transformers model (BERT) Devlin et al. (2019), associated to probabilistic models such as Conditional Random Field (CRF) Lafferty et al. (2001). This combination enables to increase the global prediction coherence with restricted fine-tuning.

Language models are essential to incorporate statistical knowledge without having to train on huge datasets. However, these models were recently criticized for their lack of transparency on how training data are reused in predictions Bender et al. (2021). Thus, the creation of collaborations between linguists and data researchers into collaborative project Gauthier et al. (2020) are an actual research topic which aims to create more linguistic oriented language models.

In this work, we study the impact of incorporating grammatical structure into the BERT-CRF model. We focus on the constituency tree representation of the sentence. This representation is composed of internal nodes (also called NT for Non Terminal) representing the grammatical structure and leaf nodes (nodes without children) corresponding to the sentence words.

The main goals of this paper are

- to evaluate GNNs models combined with a CRF layer in order to use the syntactic information contained in the constituency tree representation; [1]

---

[1] The code will be publicly available on GitHub after the blind review process. It contains a download script of the dataset, the model code, the interpretability models and the hyper parameter optimization scripts.

- to use interpretability models to evaluate the contribution of the tree structure to the final results.

## 2 RELATED WORKS

In this section, related works about Argument Mining and tree structure representation in NLP, Graph Neural Network and Conditional Random Field are reviewed.

### ARGUMENT MINING

The precise definition of the concept of argument is an important step when creating dataset annotation rules. A previous work on Argument Mining Lawrence & Reed (2020) has identified two main scales of argument units: Elementary Discourse Units (EDUs), which are the non-overlapping text spans corresponding to the atomic discourse units, and Argumentative Discourse Units (ADUs), defined as minimal analysis units. As explained in Peldszus & Stede (2013), "when two EDUs are joined by some coherence relation that is irrelevant for argumentation, the resulting complex might be the better ADU". With this definition, one can notice that EDUs and ADUs are not defined at the sentence level. Thus it is more relevant to use a dataset labeled at the token level instead of sentence level. The current biggest dataset with argument unit ADUs labels at the token level is AURC [2] Trautmann et al. (2020).

### TREE STRUCTURE REPRESENTATION IN NLP

The goal of using a tree representation of the sentence is to exploit the syntactic hierarchical dependencies of the sentence with graph neural network. Thus, the constituency tree representation is more appropriate than POS taggers which provide one tag per token (according to the Universal POS tags categories). We prefer the constituency parsing over the dependency parsing because dependency grammars phrasal constituents and phrase-structure rules do not play a direct role in dependency parsing. Thus, the distance (in terms of number of edges of the tree) between words of the same grammatical class is longer in the dependency parsing of the sentence.

Pioneer work (Gildea & Palmer, 2002) presenting the benefits of using constituency tree representation has failed to scale into production. This is caused by the lack of a reliable model to generate constituency tree representation of the sentences. However, recent promising results were made in consistency and dependency parsing. Zhang (2020)

For our preprocessing step we used a neural network model called Berkeley Neural Parser (BENEPAR) (Kitaev & Klein, 2018), which is trained on 11 languages. We use this model with the weights provided by these designers. Other possibilities for this task are the StanfordParser with Stanza Qi et al. (2020) or NLTK . We chose BENEPAR because it is integrated with Spacy and it can be used in 11 different languages.

Other papers have recently studied the use of tree structure to incorporate syntactic information to their models. Marcheggiani & Titov (2017) uses Graph Convolutional Networks (GCNs) based on the dependency tree structure of the sentence for semantic role labeling. Beck et al. (2018) uses GNNs for generation tasks from abstract meaning representation.

### GRAPH NEURAL NETWORK

Graph Neural Networks refer to the various methods which extend Deep Learning to data organized in graphs. GNNs were introduced by Gori et al. (2005) as an extension of Recursive Neural Networks (RNN). GNNs use a neural network to propagate information within the dataset geometrical structure.

In this article, the Graph ATtention (GAT) layer Veličković et al. (2018) which appears to be the most adapted message passing layer to propagate information following a self-attention strategy is used. The attention mechanism has been widely used in NLP tasks since its good results in translation tasks presented in Vaswani et al. (2017). The main benefits of attention mechanisms are that they

---

[2]The dataset is publicly available at `https://github.com/trtm/AURC`

allow to deal with variable size inputs and enable to specify different weights to different neighbours focusing on the most relevant parts of the input to make decisions.

### Conditional Random Field

Conditional Random Fields were introduced by Hammersley & Clifford (1971) as probabilistic models for magnetic field modeling. Their goal was to exploit the Markov assumption of a lattice to derive the log probability of the label distribution formula. Since then, CRFs have been widely used in machine learning thanks to the iterative scaling algorithms presented by Lafferty et al. (2001) to find the parameter that maximizes the log-likelihood of the training data. Many interactions have been made between CRFs and GNNs. Gao et al. (2019) proposed a new CRF layer for GCNs. Cui & Zhang (2019) proposed a neural network alternative to CRFs motivated by the fact that the Markov assumption is not always respected for linear chain representation of the sentence.

## 3 Model Presentation

### 3.1 BERT & CRF

This model has been introduced by Trautmann et al. (2020). It is composed of two steps. In the first step, the sentence is tokenized following the BERT tokenizer and the BERT model is fine-tuned for token classification, where the output of the last layer matches the number of classes of the dataset. In the second step, a linear chain Conditional Random Field is applied to estimate the probability of each label class from BERT.

The main idea of this model is to leverage the BERT attention knowledge and then to improve the results by incorporating a linear chain dependency structure. This takes advantage of neighbours dependency relations between words.

The good results of this model lead us to use it as a competitive benchmark for our approach based on constituency trees as input representations of sentences.

### 3.2 BERT, GNN & CRF

A major difficulty in choosing a graph neural layer is that each sentence has a different tree representation. Hence, the model needs to be agnostic to the lack of completeness of the tree structures from the training set. The message passing design enables to share the model weights among the network nodes, thus the results do not depend on the upfront global tree structure access. It also enables to deal with directed and undirected graphs as the information flow is conditioned to the existence of a directed edge between two nodes.

### Graph Attention Layer

The Graph Attention Layer Veličković et al. (2018) allows to combine the attention mechanism with the graph structure, preserving the syntactic structural information of the sentence. In order to improve the model stability, adding multi-head attention layers might be beneficial to the training step. The different heads are then aggregated in order to provide the next hidden states of the neural network. As opposed to Graph Convolution Layer, this layer allows to assign different importances to nodes of a same neighbourhood. Furthermore, analyzing the learned attentional weights may benefit to the model interpretability, as we can extract information from attention weight matrices.

In order to take advantage of the dependency structure of the sentence, we incorporate a GNN between the use of BERT and the linear chain CRF.

The idea behind this architecture is the following. First, the BERT language model outputs the sentence hidden representation. Next, the information is spread to the graph neighbours at each iteration. In that way, we expect to reach a better consistency between neighbour nodes when we train on a restricted dataset.

# 4 EXPERIMENTAL SETUP

## 4.1 DATA SOURCE AND ANNOTATION RULES

The dataset used in our study has been introduced by Trautmann et al. (2020). This dataset appears to be one of the few datasets available that annotates argument unit at the token level. It is composed of 8000 sentences, equally divided into 8 topics, and can be used for the task of argument unit identification as well as argument polarity identification.

The authors distinguished between PRO (supporting), CON (opposing) arguments and NON (non-argumentative) words.

In the cross domain setting, the topics in the train dataset are not presented in the test dataset. In the in domain setting, the topics in the train dataset are also present (with other test sentences) inside the test dataset. This enables to evaluate how important the domain knowledge are when identifying ADU. In our preprocessing step, we need to label the internal nodes NT of the constituency tree in order to learn their representation. As these annotations are not provided by the dataset, we have chosen to annotate the internal nodes with the same labeling rules as the one described above for the sentences labeling. This enables to preserve the internal logic of the labeling process.

## 4.2 CONSTITUENCY TREE CONSTRUCTION

One of the main advantages of adding a constituency tree to classical methods is the greater proximity of the words which belong to the same grammatical class than the words which only are neighbours in a linear representation of the sentence. We can also illustrate this fact with the constituency tree presented in Figure 1. For this sentence, it can be seen that the ADU units are distributed following the grammatical structure of the sentence. For instance, the words "inflation" and "the" are neighbours in the sentence but are further in the constituency tree structure. This leads to a better identification of boundaries between ADUs.

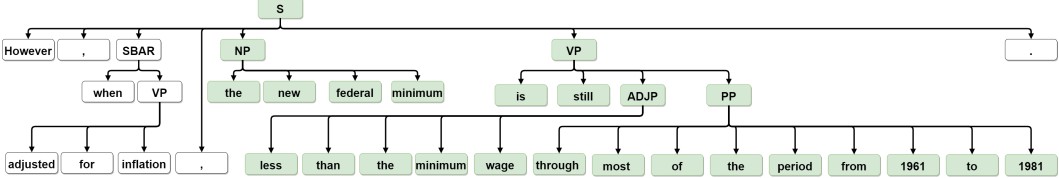

Figure 1: Constituency tree representation of the sentence, according to the Universal POS tags categories (with a maximum depth of 3): *"However, when adjusted for inflation, the new federal minimum is still less than the minimum wage through most of the period from 1961 to 1981."* about the topic "minimum wage". The green nodes represents words or spans with "PRO" label and the grey nodes represents words and spans with "NON" label.

For the construction of the constituency tree, we use Berkeley Neural Parser (BENEPAR) Kitaev & Klein (2018). This is a multilingual constituency parser which benefits from unsupervised pre-training across a variety of languages. This model provides the weights for 12 languages and a simple API incorporated to Spacy, and is thus a good choice in order to have a good accuracy, integrated into a larger pipeline. This additional pre-processing step requires to have an exact match between the tokenizations used by BENEPAR and BERT models. This constraint imposed us to remove the unmatched sentences. In addition, the samples from the in-domain test set were also excluded in the cross-domain train and dev sets. As a result, there were 4000 samples reduced to 3960 in train, 800 reduced to 790 in dev and 2000 in test reduced to 1959 for the cross-domain split; and 4200 reduced to 4157 in train, 600 reduced to 593 in dev and 1200 reduced to 1159 in test for the in-domain split. At the end, less than 2 % of the dataset were removed.

| Maximal depth of the tree / Category | No tree | 2 | 3 | 4 |
|---|---|---|---|---|
| Neighbour similarity | 97.1 % | 98.3% | 98.4% | **98.6%** |
| Similarity of grammatical classes and words | // | 90.2% | **91.3%** | 84.2% |
| Similarity of grammatical classes | // | 88.5% | **93.3%** | 92.9% |

Table 1: Evaluation of three proportions to assess the appropriateness of using the constituency tree representation. The computation was performed on the train dataset under the cross-domain setting.

## 4.3 EVALUATION OF THE NODE SIMILARITY

One of the main advantages of adding a constituency tree to argument identification methods is the greater proximity of words that belong to the same grammatical class. Indeed, we noticed that words belonging to the same grammatical class have more often the same label than words that are only neighbours in the sense of the linear representation of the sentence.

However, when the tree is too deep, the distance between words of the same grammatical class might be longer than between words of different grammatical classes. This lead us to cap the maximal tree depth allowed.

To measure this fact, we compute three statistics about the proximity of the nodes inside a constituency tree and we distinguish the results regarding to the maximal depth accepted for the constituency tree. The three proportions computed, summarized in Table 1, are the following.

- **Neighbour similarity**: the proportion of nodes which have the same label and are neighbours in the linear chain sentence representation. For the cases (columns 2, 3, 4) where we have also a constituency tree representation, we additionally restrict this set to the nodes which share the same grammatical class (i.e. which have the same parent node).

- **Grammatical class and word similarity**: the proportion of leaf nodes which have the same label as their grammatical class.

- **Grammatical class similarity**: the proportion of internal nodes which are linked by an edge and share the same label.

Table 1 shows a stronger proximity between neighbour words in the same grammatical class than neighbours words when the grammatical structure is not taken into account. We can also notice that the constituency tree with a maximal depth of 4 shows a stronger node similarity than the two others. However, 4 is not necessarily the best depth as the constituency tree with a maximal depth of 3 shows better results concerning the grammatical class similarity. This lead us to try our models with the maximum depth of 3 and 4.

## 4.4 EVALUATION SCORE

In this study, we use the F1-score as the evaluation metric of the models. The F1-score refers to the equally weighted harmonic mean of the precision and enables to take into account the precision and the recall.

There are two ways for computing the F1-score. The micro average calculates metrics globally by counting the total true positives, false negatives and false positives. The macro average calculates metrics for each label and finds their unweighted mean. This does not take label imbalance into account. Therefore, the micro F1-score is preferable for our studies as the dataset is class-imbalanced regarding the non-argument label NON at the token level.

The F1-score is computed at the token level of the BERT token. Since some words are split into two subtokens with the BERT tokenizer (in order to separate suffixes), accuracy and recall evaluations are slightly biased. One way to minimize this effect is to merge the labels of the subtokens of a word that has been split.

## 5 MODELS RESULTS

### 5.1 HYPERPARAMETERS OPTIMIZATION

The BERT-GAT-CRF model have many more hyperparameters than the BERT-CRF model. This is due to the large amount of hyperparameters of a GAT network: number of layer, units per layer, number of heads.

In order to choose the best hyperparameters for this model, we use the library Optuna (Akiba et al., 2019). Optuna is a framework for cost-effectiveness of hyperparameters optimization. The two main advantages of Optuna for our study are the following.

- Searching strategy: Optuna implements a relational sampling method that can identify the underlying concurrence relations after some independent samplings.
- Pruning strategy: regarding the intermediate F1 score values, Optuna interrupts the non promising trials based on the Successive Halving algorithm Li et al. (2020), which continues the trials if only the actual F1-scores is among the best intermediate results.

The importance of the different hyperparameters of our model has been evaluated in Table 2. It can be observed that the most important hyperparameters are the learning rate and the maximum allowed value for the gradient. From an empirical point of view, we notice that when we do not constrain the gradient, the model converges to a local optimum which is the assignment of the label NO to each word. This local optimum exists because the dataset is unbalanced towards the absence of an argument.

|  | Test Intervals | Best values | Parameters importance |
|---|---|---|---|
| Learning rate | $10^{-5}$ to $10^{-3}$ | $2.8 \cdot 10^{-5}$ | 30 % |
| Maximum gradient allowed | $10^{-1}$ to $10^{2}$ | 9.7 | 49 % |
| Number of GAT layers | 1 to 3 | 2 | 2 % |
| Number of unit per GAT layers | 50 to 300 | 290 and 100 | 2 % |
| Number of heads per GAT layers | 1 to 3 | 3 and 3 | 7 % |
| Number of linear layers | 1 to 3 | 2 | 5 % |
| Number of unit per linear layers | 50 to 250 | 100 and 100 | 5 % |

Table 2: Feature importance of the BERT-GAT-CRF model in Cross Domain prediction

One difficulty encountered during the development of this model was the management of batches. In Pytorch Geometric, we associate a graph with each input. This graph is represented by a two-dimensional matrix that contains the edges between the nodes. Thus, in order to allow different sizes of graphs, the batches matrix used in Pytorch is unfolded into a list and a list of indices links the input data to the batch indices. Thus, as we have to switch between the two representations in order to use BERT and a GNNs within our model, this reduces the computational speed.

The global training step requires loading the BERT model and a batch of data into memory. According to our implementation, this requires 30 GB of memory (GPU or CPU). We therefore performed our global training experiments on the CPU, which took about 2 hours per iteration and led to about 30 iterations per model configuration. The transfer learning model training required about 7 GB and could therefore be run on the GPU. We therefore had about 200 trials per model hyper-parameter optimisation.

### 5.2 MODELS EVALUATION

Consider first the performance of the models when the BERT model is not fine tuned. In this case, the model BERT + CRF shows poor result: 31% of F1 score probably due to the low number of parameters (we only train here the last layer of BERT and the parameters of the CRF). The BERT+GNN+CRF model reaches 50% of F1 score on the Cross Domain dataset.

In a second step, we performed a global training of the models. In the initial paper from Trautmann et al. (2020), the following metrics were presented : token level, span level and sentence level. Our model focuses on improving the argument border recognition and not the argument stance

| | In Domain | Cross Domain |
|---|---|---|
| BERT large | 68 % * | 59 % * |
| BERT large - Linear chain CRF | 69 % * | 61 % * |
| BERT large - GAT - CRF - Maximum depth allowed for Constituency tree is 3 | 72.8 % | 68.2 % |
| BERT large - GAT - CRF - Maximum depth allowed for Constituency tree is 4 | **73.5** % | **68.7** % |

Table 3: F1- score of the different models at token level on the test dataset (the results with a star (*) have been directly taken from (Trautmann et al., 2020))

identification. Thus, it improves the state of the art on the token level (but achieves similar results on the sentence level and the span level).

The results corresponding to the token level classification are presented in Table 3. As suggested by Table 1, we computed our models with a maximum depth of 3 and 4. We reached our best results with the model composed of BERT-GNN-CRF when we allowed a maximal depth of 4. This achieves 74.1 % on token level F1 score micro in a cross-domain setting and 73.8 % on token level F1 score micro for the in-domain setup.

## 5.3 MODEL INTERPRETABILITY

In this subsection, we study the interpretability of the model BERT-GNN-CRF.

Consider the problem of attributing the model prediction to its input features. First, we use the Integrated Gradient method (Sundararajan et al., 2017) which computes a score for every input of the model compared to a baseline value. The score is calculated by integrating the gradient along the input representation from the baseline to our input. When this gradient is higher, our input data is more relevant for prediction. In the BERT model, this baseline is the token "[PAD]" (commonly used for padding) which indicates the absence of a word.

We used the Captum framework (Kokhlikyan et al., 2020) which implements a wide variety of interpretability models. Figure 2 presents the contribution of the words of the sentence to the final output. In this example, it can be seen that the word contributions follow the grammatical structure (expect for the word "then", due to the fact that labeling did not follow strictly the grammar).

Figure 2: Word Importance of the Integrated Gradient Method for the sentence : *"If you have a fear of losing your important data to hard drive failure, then you can use this drive clone tool to clone your mac hard drive or volume and restore the lost data, in case of media failure."*. The "NT" token refers to the sentence internal nodes. The words highlighted in green (resp. red) have a positive (resp. negative) impact on the final outputs.

Then, we study the importance of the edges for the model prediction. The Integrated Gradient method cannot be directly applied to our graph study as we cannot integrate through the edges weights (the edges are not weighted). We compute a simple algorithm to measure the importance of the different edges of our graph. We iteratively hide one edge from the graph and compute the model to see how the result varies. By doing so with all the edges we can measure the necessity of every edges taken independently. The algorithm is described in the Algorithm 1.

The results of the interpretability on edges can be found in Figure 3. It globally shows that the words which are labeled with an argument (in color) furnish more information to their related edges. This confirm the utility of adding the constituency tree representation to our model.

---

**Algorithm 1** Measure of the impact of the graph edges.

---

**Input**: A sentence to analyse, The edges of the constituency tree of the sentence, The labels of the sentence

**Output**: matrix containing the importance weight of each edge

1:  Run the model on the input sentence
2:  **for** Every edges of the graph **do**
3:      Remove one edge of the graph
4:      Run the model with the new input
5:      Compute difference between the previous loss and the new loss
6:      Store this value as the importance of the edge
7:  **end for**

---

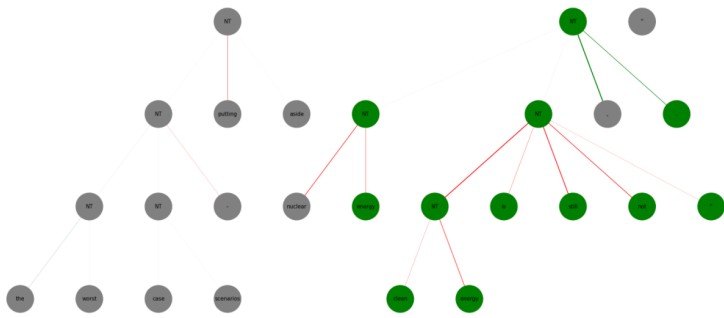

Figure 3: Tree representation of the edges interpretability methods. The red edges refer to edges which have a positive impact on the prediction. The green edges refer to edges which have a negative impact on the prediction. (This figure has been generated by NetworkX)

## 6    CONCLUSION

In this study, we established a new method for identifying the ADU boundary using the sentence constituent tree representation. This model allows the information to be spread across the graph and performs well on a small dataset.

The two main errors previously observed on this dataset were:

- the span of an argumentative segments was not correctly recognized.
- the stances are not correctly classified.

In this paper, we focus on the span detection problem and improve the method for identifying the boundaries of an ADU. However, we do not have a strict adherence to grammatical correctness. As mentioned in Trautmann et al. (2020), this would require that all spans containing arguments be clauses. There are examples of sentences where subjects are absent, which limits the performance of models using grammatical structure. This problem could be solved by specifying annotation rules that more strictly follow the syntactic structure of the sentence.

Regarding the second type of error, the problem of position identification is mainly related to the performance of the BERT model. The dataset only provides sentences with a maximum length of 64. This implies that only a narrow context of the ADU is available, which limits the capability of our model. Many arguments require more domain knowledge to understand what is at stake.

They model arguments as self-contained pieces of information that can be verified as relevant arguments for a given topic with no or minimal surrounding context.

A disadvantage of using message-passing GNNs is that, since not all words have the same depth in tree representations, message passing is treated identically across words. A word with shallow depth will pass information to a much more distant part of the tree than a node with greater depth.

In future work, we will explore other GNNs to overcome this problem. We plan to continue our study on argument recognition by working on datasets with more surrounding context.

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
