# OpenReview forum: "Constituency Tree Representation for Argument Unit Recognition"
_ICLR.cc/2022/Conference — ICLR 2022 Submitted_

### Official Review · Reviewer_cyhs · 2021-11-01

**Correctness:** 2
**Technical Novelty And Significance:** 2
**Empirical Novelty And Significance:** 2
**Recommendation:** 3
**Confidence:** 4

**Main Review:**


Some comments:

The concept of Argument from Sentences is not properly described until Section 2, first paragraph. The entire abstract and introduction you are wondering what the paper will be about. Also, the description of Argument mining will be much better with an example.
the abstract starts too ambiguous. It is not even clear what "arguments" from sentences are and why neighboring words are relevant.

- "However, these models were recently criticized for their lack of transparency on how training data are reused in predictions."
This claim requires a citation.

- "However, these models were recently criticized for their lack of transparency on how training data are reused in predictions. Thus, model architectures based on linguistic concepts seem o be needed to improve current models and increase their explainability. "
Why are you reaching this conclusion? The two sentences seems a bit disconnected.

- "the most commonly used models do not do not take into account the grammatical structure of sentences, which leads to a gap between current NLP models and linguistic analysis."
There is in fact a lot of work on showing how NLP models are able to understand linguistic structure, this is one place to start: https://aclanthology.org/N19-1419.pdf

-"This is caused by the lack of a reliable model to generate constituency tree representation of the sentences. However, recent promising results were made in consistency and dependency parsing"
Again, these require citations. The first statement and the second one, both of them.

The related work section contains technical details about your experimental setup. It is hard to follow.

"The attention mechanism was introduced by Vaswani et al. (2017) for translation tasks in NLP and has since been widely used in many other tasks. " The attention mechanism was not introduced by Vaswani et al. (2017), I would recommend to thoroughly read Vaswani et al. (2017) to find out more.  --> Dzmitry Bahdanau, Kyunghyun Cho, and Yoshua Bengio. Neural machine translation by jointly
learning to align and translate. CoRR, abs/1409.0473, 2014.

There are many more like this, please make sure that when you make a claim that is not supported by your own results, you cite relevant work, also make sure to find the appropriate paper(s) to cite for relevant concepts.





**Summary Of The Paper:**

This paper presents how to extract arguments (Elementary Discourse Units (EDUs), which are the non-overlapping text spans corresponding to the atomic discourse units, and Argumentative Discourse Units (ADUs), defined as minimal analysis units) from sentences by incorporating constituency tree representations (using BENEPAR parser, Kitaev and Klein) in a BERT with a- Graph neural network and a subsequent CRF (explained in Sect 4.2).

The authors show that the constituency tree representations are relevant since the ADU units are distributed according to the grammatical structure of the sentence which is captured by the tree, and this is supported by empirical results.





**Summary Of The Review:**

The paper lacks many details that, in my opinion, preclude it from publication at this stage. It makes bold claims that are not supported by citations or results (see Main review section).

The results show improvements over a BERT model that does not use the syntactic information nor CRF. I would recommend the authors to train multiple models (e.g. 5) with different random seeds and show average scores and standard deviation, it would help to ensure that your improvements are not just for the initialization you chose.

---

> ### Author Response · Authors · 2021-11-19
> **Answer to your questions**
>
> Dear reviewer, thank you for your time and your accurate analysis.
>
> We have published a new version of the article taking into account the remarks of all the reviewers.
>
> Please find below the answers to your questions numbered according to your questions.
>
> 1.	We added a definition of an argument in the introduction.
> 2.	Thank you for noticing the issue regarding the citations ; we improved the accuracy of the bibliography.
> 3.	We compared our results with the baseline composed of BERT and CRF.

---

> > ### Comment · Reviewer_cyhs · 2021-11-20
> > **checking new version uploaded by the authors**
> >
> > The bibliography has improved in the new version, but many of my comments about it are still ignored. For example, Vaswani et al. 2017 is still cited as the first occurrence of attention mechanism in the literature.
> >
> > The authors added a sentence at the top of the intro about argument but it is not connected with the rest and it does not help to improve the coherence of the story.

---

### Official Review · Reviewer_iNq3 · 2021-11-01

**Correctness:** 3
**Technical Novelty And Significance:** 2
**Empirical Novelty And Significance:** 2
**Recommendation:** 3
**Confidence:** 5

**Main Review:**

Strengths

The paper presents a simple method for incorporating syntactic information for the task of ADU recognition. The hyperparameters are well documented, and strong empirical results.

Interesting analysis for understanding which substructure of constituency tree is important.

Weaknesses

The main weakness is that the relationship between ADU and constituency tree is not clearly described. Is it true that ADU are often phrases that occur in the constituency tree? How often does this happen? Does the new BERT-based model adhere to constituency constraints? Is the BENEPAR parser appropriate for this data? Based on Trautmann et al.’s comments on grammaticality and clauses, my intuition is that an ADU is almost always a phrase in the tree, in which case it is somewhat less surprising that it helps this task (and maybe it should be helping even more). If this is the case, it’s worth considering related work in distant supervision to include.

The treatment of constituency trees is haphazard. 1) At times it is not clear if dependency or constituency trees are being used; 2) Table 1 should be made more clear, what do the percentage values indicate, and why not use the full tree at all; 3) It is worth adding to the related work more work in constituency tree representation, such as but not limited to Yang and Deng 2020 that also use GNN to represent constituency tree.

(low priority) Some technical details are concerning as described. For example, it is true that batching heterogeneous graphs may be somewhat more challenging than batching similar length sequences, but it is hard to believe this is one of the “major difficulties” of this work. If it is such a challenge, then it may have warranted further discussion about tradeoffs in architecture selection and impact on speed or performance.


**Summary Of The Paper:**

The authors present results using syntactic information (primarily through constituency trees) on the task of recognizing argument discourse units (abbreviated to ADU). Their results are strong the paper is clearly lacking in multiple dimensions: a) it is not made clear the similarities between ADUs and constituency trees, and the lack of this clarity makes the research story less compelling; b) in general, the paper is difficult to follow (e.g. it is not clear whether dependency structure or constituency, or both, are being used in the model); and c) the paper does not present any new methods, primarily relying on the previously published GAT model --- although, the lack of novelty is not the sole issue with this work. For these reasons, I do not recommend to accept this paper at this time.


**Summary Of The Review:**

I recommend to reject this paper for the following reasons: a) it does not sufficiently motivate the use of syntax, and does not make it clear relation between ADU and constituency; b) it is haphazard in its use and description of its main feature, constituency trees; c) the writing is lacking, it does not tell a particularly compelling research story, nor cover related work particularly well. In summary, it is not particularly strong in any dimension, and the main result does beat the baseline, but can easily be explained by the connection between constituency and ADU.

---

> ### Author Response · Authors · 2021-11-19
> **Answer to your questions**
>
> Dear reviewer, thank you for your time and your accurate analysis.
> We have published a new version of the article taking into account the remarks of all the reviewers. Please find below the answers to your questions numbered according to your questions.
>
> 1.	Is it true that ADU are often phrases that occur in the constituency tree? How often does this happen?
>
> ·	The goal of the table 1 is to answer these questions. We computed the frequency of words belonging to the same grammatical classes and having the same labels. Please see Section 4.3 and Table 1 for more details.
>
> 2.	Does the new BERT-based model adhere to constituency constraints?
>
> ·	We used BERT model as a language model which gives us an embedding of the sentence. We motivated this choice to preserve a model comparable to the previous model proposed by Trautmann. We acknowledge that BERT might not be the best language model for argument mining, and we are studying other language model for our future works.
>
> 3.	Is the BENEPAR parser appropriate for this data?
>
> ·	We chose BENEPAR mainly for its good performances and since it is easy to incorporate it into a Spacy pipeline.
>
> 4.	At times it is not clear if dependency or constituency trees are being used
>
> ·	We precised that we are only using constituency parsing in Section 2.
>
> 5.	Table 1 should be made more clear, what do the percentage values indicate, and why not use the full tree at all
>
> ·	We added more explanations concerning Table 1.
>
> 6.	For example, it is true that batching heterogeneous graphs may be somewhat more challenging than batching similar length sequences, but it is hard to believe this is one of the “major difficulties” of this work.
>
> ·	The issue is the compatibility between pytorch geometric models and pytorch models. We rephrased this as “One difficulty” to avoid confusing the reader.

---

> > ### Comment · Reviewer_iNq3 · 2021-11-28
> > **n/a**
> >
> > Thank you for the response. Table 1 remains quite difficult to understand, but more importantly, is not a particularly compelling motivation for constituent labels, plus, it appears that constituent labels are being re-purposed as word labels which is quite unusual. For example of a motivation commonly used in other works, constituent trees are often helpful for disambiguating multiple valid interpretations of a sentence, or for identifying a few of the more salient phrases, albeit this is highly dependent on the accuracy of the parser and the parser's fit for the current domain (a parser trained on newswire may or may not work well on other types of data).
> >
> > Overall, I think my initial concerns still hold and I keep my score unchanged.

---

### Official Review · Reviewer_ExWX · 2021-11-01

**Correctness:** 2
**Technical Novelty And Significance:** 2
**Empirical Novelty And Significance:** 2
**Recommendation:** 3
**Confidence:** 4

**Main Review:**

Strengths:
1. This paper proposes a novel method in addressing argument discourse unit detection, by incorporating constituency parser output via Graph Attention Layer. Although the reported results are better than the baseline, the scores are not comparable. Unlike previous works (Trautmann et al., 2020), the paper also does not report the complete result (see weaknesses below).

Weaknesses:
1. This paper has a poor literature review, motivation, and presentation, making it hard to read. For instance, a) In Section 1, many references are missing for argument analysis definition and examples, previous works in par3, and reference to the SPMRL dataset. Par4 is also confusing. The motivation to incorporate syntax into this task is also very weak. I suggest providing an example to show why syntax can be a good feature for ADU detection; b) In Section 2, par1-2 have a bad flow; par4 mentions “recent promising results”, but which one? It needs some references. c) It’s strange to have a Section of Experimental set-up prior to model architecture. Section 4.1 also has a bad naming (I suggest renaming it as the model name); d) The proposed method needs mathematical/formal definition which is completely missing in this paper.
2. Although the dataset is based on Trautmann et al., 2020, the paper does not clearly state the original train/dev/test split of data, as well as general statistics of stance labels. I don't think the reader needs to check the original paper for this basic information.
3. The motivation for pruning the constituency tree is not properly presented since Section 1. For instance, a) in Figure 1, why is the depth of the tree capped to a maximal value of three? b) in Section 3.2, “Maximum depth” is not clearly explained and poorly presented. What does the author mean by “columns 2, 3,4”? Which data partition is used to determine optimal “maximum depth”? It sounds to me that this value is determined based on all the data, which potentially harms the methodology.
4. Unclear motivation and setting of using BENEPAR constituency parser. The authors mention that BENEPAR is a multilingual parser that can parse French and English sentences, however, in the reported result, there is no experimental result for both languages. Instead, the results seem to be in 1 language.
5. The authors discard 1300 samples of data due to tokenization requirements of BERT and BANEPAR, which I think is a very strange motivation. Moreover, the main experimental result in Table 3 uses baseline results from Trautmann et al., 2020, which can be misleading as the data is not comparable anymore.
6. The reported result does not have a clear description of in-domain and cross-domain data, nor further explanation or analysis.
7. This paper does not report the full result of experiments, only token-level F1-score. In Trautmann et al., 2020, there are three different metrics: token-level, segment-level, and sentence-level

**Summary Of The Paper:**

This paper demonstrates the utility of constituency parser for improving argumentative discourse unit (ADU) detection. ADU is defined as a “minimal unit of analysis” in argument mining and the task is to first segment the ADU spans, and then predict the stance label. This paper uses Trautmann et al., 2020 dataset that comprises 8000 sentences with 3 stance labels: PRO (supporting), CON (opposing), and NON (non-argumentative). The paper alters the architecture of BERT-CRF in the previous works, by adding Graph Attention Layer (GAT) prior to CRF. GAT is computed based on pruned constituency trees, yielding improved performance compared to BERT-Large.

**Summary Of The Review:**

Although the paper has a novel and interesting idea, I argue it is not ready for publication due to the aforementioned weaknesses above. The critical problems are incomparable results with baseline as well as its incompleteness issues.

---

> ### Author Response · Authors · 2021-11-19
> **Answer to your questions**
>
> Dear reviewer, thank you for your time and your accurate analysis.
> We have published a new version of the article taking into account the remarks of all the reviewers.
> Please find below the answers to your questions numbered according to your questions.
>
>
> 1.
> a.	One objective on Figure 1 is to give a visual motivation about why using Constituency Tree Representation.
>
> b.	We added the missing reference about “recent promising results” in Section 2.
>
> c.	Thank you for noticing, we switched the two sections.
>
> d.	The goal of this paper is not to provide a complete theoretical motivation about the method. We are working on a more theoretical paper with strong evidence on the link between ADU and different models related to Conditional Random Field.
>
> 2.	We added a few lines describing in details the train, test, dev split. (Section 4.2)
>
> 3.
> a.	We changed the figure 1 to present the total constituency tree representation.
>
> b.	We added a few lines to explain why we needed to prune the constituency tree. (Section 4.3). We precised that the statistics were performed on the train dataset under the cross-domain setting. (In the title of Table 1)
>
> 4.	We chose BENEPAR mainly for its good performance and since it is easy to incorporate into a Spacy pipeline.
>
> 5.	We admit that this part lacked explanation. We detailed the modifications on the dataset. This lead to a 1.6% reduction. (Section 4.2)
>
> 6.	We did not want to paraphrase the original paper which introduced the dataset. We add a few lines describing it in Section 4.1.
>
> 7.	The goal of our model is to improve the border identification of argument components. At the sentence and span level, our model processes as the original model. We added a sentence in order to precise that we do not improve the model on other metrics than the one we presented. (Section 5.2)

---

### Official Review · Reviewer_sFFn · 2021-11-03

**Correctness:** 3
**Technical Novelty And Significance:** 2
**Empirical Novelty And Significance:** 2
**Recommendation:** 5
**Confidence:** 4

**Main Review:**

The method used in this paper is more alike a connection of three existing methods, bert, gat, and crf. Currently, could find little novel methods other than bert-gat-crf in this paper. The target questions is rather limited to a specific NLP task of semantic parsing field. Also semantic role labeling and related baselines/datasets should be involved for some direct comparison of this “argument unit recognition” task.




Detailed questions and comments:
1.	Figure 1, why use 0 to 6 as higher level nodes? I think using POS-tags and phrase-tags are more readable, such as article tag for “The”, verb tag for “is” and noun phrase for “the time”.
2.	Prefer to see detailed examples that are improved by this bert-gat-crf method for intuitive understanding.
3.	Some cites can be improved, such as the bottom lines in page 5. With Roberta liu et al. (2019)? -> Roberta (liu et al. 2019).
4.	What are the reasons of not using dependency trees and give a comparison as well for this task? Dependency tree structure are shallow parsing and can be understood easily and adapted for semantic role labeling of argument unit detection and also predicate-argument structure prediction as well.


**Summary Of The Paper:**


This paper presents a bert-gat-crf framework for recognizing argument units in sentences through constituency tree representation. Examples of what is a consistency tree of short and long sequences are given. Experiments on small datasets proved the performance of the bert-gat-crf algorithm.


**Summary Of The Review:**

Strong:
1.	Using of bert-gat-crf for argument unit detection and achieved improvements in small-scale datsets;

Weak:
1.	The method of bert-gat-crf is less novel for this paper;
2.	The target question is a pure NLP task and also related to other tasks such as semantic role labeling that better mention and compare with their baselines as well.

---

> ### Author Response · Authors · 2021-11-19
> **Answer to your questions**
>
> Dear reviewer, thank you for your time and your accurate analysis.
> We have published a new version of the article taking into account the remarks of all the reviewers. Please find below the answers to your questions numbered according to your questions.
>
> 1.	Figure 1, why use 0 to 6 as higher-level nodes? I think using POS-tags and phrase-tags are more readable, such as article tag for “The”, verb tag for “is” and noun phrase for “the time”.
>
> We chose initially to replace the internal nodes by numbers in order to highlight the fact that we do not consider the grammatical class into this model but only the tree structure. In the revised version, we followed you recommendation and let appear the tags.
>
> 2.	Prefer to see detailed examples that are improved by this bert-gat-crf method for intuitive understanding.
>
> Figure 2 already illustrates the necessity of the tree structure.
>
> 3.	Some cites can be improved, such as the bottom lines in page 5. With Roberta liu et al. (2019) ? -> Roberta (liu et al. 2019).
>
> Thanks for noticing, we changed it.
>
> 4.	What are the reasons of not using dependency trees and give a comparison as well for this task? Dependency tree structure are shallow parsing and can be understood easily and adapted for semantic role labelling of argument unit detection and predicate-argument structure prediction as well.
>
> We used at first the dependency tree and we did not achieve good results. We finally chose to focus our attention on constituency parsing as we think the ADU are strongly connected to constituents which are only indirectly represented in the dependency structure.

---

### Decision · Program_Chairs · 2022-01-20

**Decision:**

Reject

**Comment:**

The paper present results using syntactic information (primarily through constituency trees) on the task of recognizing argument discourse units. No reviewer recommends acceptance of the paper:
- The empirical results appear strong, though the reviewers raise questions about some of the experimental choices.
- The writing is unclear and reviewers point out many missing or incorrect references in the bibliography.
- There is little methodological novelty - known techniques are applied to a topic that has not been studied much.
Overall, the area chair agrees with the reviewers that this work does not yet meet the bar for ICLR.